# Advances and Progress in Self-Healing Hydrogel and Its Application in Regenerative Medicine

**DOI:** 10.3390/ma16031215

**Published:** 2023-01-31

**Authors:** Wei Zhu, Jinyi Zhang, Zhanqi Wei, Baozhong Zhang, Xisheng Weng

**Affiliations:** 1Department of Orthopaedics, Peking Union Medical College Hospital, Chinese Academy of Medical Sciences & Peking Union Medical College, Beijing 100730, China; 2School of Medicine, Tsinghua University, Beijing 100084, China

**Keywords:** self-healing hydrogels, regenerative medicine, biomaterials

## Abstract

A hydrogel is a three-dimensional structure that holds plenty of water, but brittleness largely limits its application. Self-healing hydrogels, a new type of hydrogel that can be repaired by itself after external damage, have exhibited better fatigue resistance, reusability, hydrophilicity, and responsiveness to environmental stimuli. The past decade has seen rapid progress in self-healing hydrogels. Self-healing hydrogels can automatically self-repair after external damage. Different strategies have been proposed, including dynamic covalent bonds and reversible noncovalent interactions. Compared to traditional hydrogels, self-healing gels have better durability, responsiveness, and plasticity. These features allow the hydrogel to survive in harsh environments or even to be injected as a drug carrier. Here, we summarize the common strategies for designing self-healing hydrogels and their potential applications in clinical practice.

## 1. Introduction

Hydrogels are water-swollen three-dimensional polymer networks [1], formed by covalent bonds or noncovalent bonds such as hydrogen bonds and van der Waals forces [2]. The interaction between hydrogels counterbalance bonds, leading to expansion of polymer networks, which also regulates hydrogels’ internal transport, diffusion characteristics, and mechanical strength. Due to their three-dimensional structure, hydrogels are similar in structure to the extracellular matrix [3] and have good biocompatibility, excellent water retention ability, and a large specific surface area [4]. The properties of hydrogels result from their highly swollen, hydrophilic 3D cross-linked polymer network structure, mimicking hydrated extracellular matrix (ECM) and facilitating nutrient and oxygen transport due to its porous structure. Because of existing hydrophilic groups such as amine, hydroxyl, and carboxylic distributed throughout its 3D network, the hydrogels exhibited extraordinary hydrophilicity. Therefore, hydrogels have long been an exciting and promising tool in tissue engineering, wound dressing, and other biological fields [4,5].

Hydrogels can be endowed with many properties that are advantageous for biomedical applications. The non-covalent interaction and grafted bioactive group of hydrogels can improve the biocompatibility of the hydrogel. High porosity can bring a large expose surface and ensure the material exchange that supports bioactivity. In terms of pore size, the different density of the hydrogel network leads to multiple applications. Larger pores at 100μm level can provide space for vasculature, while pore size below 1 μm can effectively hinder bacteria penetration. Different surface properties can be achieved by modifying the exposed part of the hydrogel surface. For example, aldehyde groups can be used to react with local amines in cartilage tissue to achieve adhesion, while the materials in the circulatory system need to be smooth and minimize hemocyte attachment. As long as the elasticity and swelling behavior of hydrogel mimic target tissue and have appropriate surface adhesion effects and pore structure to ensure cell attachment and metabolite delivery, the hydrogel can be ideal material for regenerative medicine.

However, despite their many favorable properties, the main drawback of hydrogels is their low strength, leading to mechanical damage, which greatly shortens the lifetime and limits the application [6]. Self-healing hydrogels [7], a new type of hydrogel that can be repaired by itself after external damage, have exhibited better fatigue resistance, reusability, hydrophilicity, and responsiveness to environmental stimuli. They show great advantages in regenerative medicine compared with traditional hydrogels [8,9]. Two strategies are usually followed when designing self-healing hydrogels. One is to embed microcapsules in materials. When cracks occur and destroy the outer wall of the capsule, the polymer monomer stored in the capsule disperses to the fracture position, and the catalyst in the material causes polymerization, resulting in self-healing. The other is based on interactions between polymer chains, which are generally reversible. If the fracture is formed by external force, the material may have a self-healing property due to the interchain force at the fracture site. The self-healing properties of these materials can be repeated, so they have attracted much attention in recent years.

Polymer chains, reversible in self-healing hydrogels’ restoration, can be divided into dynamic covalent and noncovalent bonds [8,9]. Disulfide bond [10], Diels–Alder cycloaddition [11], imine key/Schiff base function [9], etc., could be adopted in self-healing as dynamic covalent bonds. Noncovalent bonds include metal ligated interactions, hydrogen bonding interactions, interactions between the subject and object, electrostatic bonding (reaction between polyanion and anion), hydrophobic associations, polymer nanocomplex interactions, etc. [9]. Among them, hydrogen bonds, metal ligated interactions, hydrophobic associations, and the combination of multiple forces are widely used in hydrogels [12].

Polymers of self-healing hydrogels for medical purposes, such as chitosan and alginate, often exhibit high biocompatibility, and the metabolites after degradation are generally harmless. The polymer backbones can be modified to create appealing features. These characteristics make hydrogels with suitable 3D structures able to support cellular activity, and their chemical properties, such as their hydrophilic nature, promote cell proliferation, adhesion, and differentiation. By adjusting the ion concentration, pH, and biological active substances involved, the hydrogel can be adjusted to adapt to different target tissues [13].

Compared with traditional hydrogels, self-healing hydrogels show promising application potential in different fields [12]. For example, hydrogels capable of self-healing at a low pH (pH = 1–5) can be used as sealants in acid-containing containers and offer great advantages in the treatment of gastric ulcers. In addition, since hydrogel inserts do not require surgical incisions, injectable self-healing hydrogels can deliver drugs to the body without causing obvious harm. Previously published reviews mainly introduced its mechanism and summarized the conductivity of self-healing hydrogel flexible sensors [7,14]. Other reviews focused on the single component, such as pectin-based or polypeptide-based self-healing hydrogels [15,16], unified compound mechanisms such as chitosan-based hydrogels [17], or cross-linking and supramolecular self-healing hydrogels [18,19], and discussed how they could be applied in a specific medical application [20,21]. In addition, there are discussions of the investigation of hydrogels with unique features such as injectability [19,22]. Our review systematically described the design strategies of different mechanisms, especially based on compounding motif bonding and components of self-healing hydrogels in recent years, and illustrated the promising clinical applications.

## 2. Self-Healing Hydrogel Design Strategy

Here, we mainly discuss strategies based on reversible interactions that can re-form after rupture (Shown in Table 1). The self-healing features of the hydrogel could be tested with compression and tension experiments including rigidity, tensile, shearing, and compressive strain, as well as self-healing at the macro scale after being cut apart. The interaction between atoms close to each other can be reestablished after the fracture, accompanied by the diffusion of materials, leading to the healing of the fracture.

### 2.1. Dynamic Covalent Bond Base Strategy

The hydrogel polymer networks with traditional covalent bonds crosslinked are irreversible and tend to show brittle properties, which make them subject to fatigue or damage. As a kind of chemical bond that can be broken reversibly in a mild environment, the dynamic covalent bond has been used in the design of self-healing hydrogels. The following are the main strategies when designing self-healing hydrogels based on reversible covalent bonds.

#### 2.1.1. Imine Key/Schiff Base Function

An imine bond is a chemical bond formed by a reversible condensation reaction of primary amine and an active carbonyl group (such as an aldehyde or ketone group), commonly known as a Schiff base bond. A Schiff base bond can be broken and recombined dynamically, and so is widely applied in hydrogel networks. The crosslinking of amino and aldehyde and ketone groups in polymers, such as chitosan and polyethylene imine, leads to reliable Schiff base bonds.

Yan et al. [45] reported a self-healing hydrogel based on dynamic Schiff base bonds; its self-healing mechanism is shown in Figure 1. The self-healing behavior of the hydrogel was realized by the dynamic Schiff base bond (–CH=N–) formed between the amino group of polyethylene imine (PEI) and the aldehyde group of dibenzaldehyde functionalized polyethylene glycol (PEG). The rheological strain scanning step test experiment proved that the hydrogel containing a Schiff base bond had excellent self-healing ability. Zhao Wei et al. [46] developed a novel self-healing hydrogel CEC-OSA-ADH based on a biocompatible polysaccharide by using the dynamic reaction of N-carboxyethyl chitosan (CEC), adipic dihydrazide (ADH), and sodium alginate (OSA). The CEC-OSA-ADH hydrogel network showed excellent self-healing ability and high healing efficiency (up to 95%) under physiological conditions due to the presence of a dynamic imine bond and hydrazone bond, without any external stimulation. The rheological recovery test further verified the excellent self-healing ability of the hydrogel. In addition, the CEC-OSA-ADH hydrogel showed good cytocompatibility and cell load and release ability, which provided the material basis for its application in the biomedical field.

#### 2.1.2. Disulfide Bond Base

Disulfide bonds are dynamically reversible covalent bonds, which are ubiquitous in organisms and play an important role in maintaining the tertiary structure of proteins and intracellular redox potentials [48,49]. This specific covalent bond can be rapidly transformed with mercaptan due to the influence of pH value and an oxidation reducing agent and is widely used in the preparation of self-healing hydrogels [50,51,52].

Barcan et al. [53] reported a self-healing hydrogel generated from triblock copolymers and a crosslinking dithiol, through an expedient organocatalytic ring-opening polymerization of cyclic carbonates containing pendant dithiolanes (trimethylene carbonate/dithiolane, TMCDT) from poly(ethylene oxide) diols to generate water-soluble triblock (ABA) copolymers containing a central poly(ethylene oxide) block and terminal dithiolane blocks. The dynamic stepwise strain experiment on the synthesized hydrogel exhibited good self-healing ability. When the strain of the hydrogel increased from 1% to 800%, the energy storage modulus G decreased from 2300 Pa to 65 Pa, and when the strain recovered to 1%, the energy storage modulus quickly recovered to the original value. Deng et al. [54] developed another self-healing hydrogel based on disulfide and acylhydrazone bonds. In alkaline environments, the hydrogel realized self-healing through the disulfide bond to mercaptan transformation reaction. The self-healing process was triggered at room temperature without the need for external stimulation. Stress–strain test curves of self-healing hydrogels for 48 h showed that the fracture stress and strain could reach more than 50% of the original strength and elongation. The presence of acylhydrazone also makes the gel self-healing under neutral and acidic conditions.

#### 2.1.3. Diels–Alder Reaction

The Diels–Alder (DA) reaction is considered the most ideal covalent bond in crosslinked hydrogels because it is fast and has high efficiency, better selectivity, and no by-products; in addition, it is reversible and can be used to prepare self-healing hydrogels. Shao et al. [55] developed a simple and effective method to prepare a CNC–PEG nanocomposite hydrogel by crosslinking furan-based modified cellulose nanocrystals (CNC) with functional poly(ethylene glycol) PEG of maleimide terminal through a reversible DA reaction. The hydrogel showed good mechanical properties, fatigue resistance, and self-healing, and its self-healing was promoted at 90 °C, with the mechanical properties recovered by 78%. The uniaxial tensile tests and unconfined compression tests displayed the outstanding mechanical properties of the hydrogels, with a high fracture elongation up to 690% and a fracture strength up to 0.3 MPa at a strain of 90%. The good self-healing performance of CNC–PEG hydrogels provides a new idea for the design of high-performance biocompatible hydrogels. Zhao et al. [56] reported the preparation of a reversible DA reaction crosslinking dextran-based self-healing hydrogels under physiological conditions (pH = 7.4, 37°) using cytocompatibility-rich fulvene-modified dextran as the main chain and dichloromaleic acid-modified poly(ethylene glycol) as the crosslinking agent. The hydrogels showed excellent self-healing properties under physiological conditions. This study provides a simple method to prepare polysaccharide self-healing hydrogels and extends the potential applications of self-healing hydrogels in biomedical fields such as cell encapsulation.

### 2.2. Nondynamic Covalent Bond Base Strategy

#### 2.2.1. Ionic Interaction

Using ionic interaction between oppositely charged groups is a common strategy to pull polymer chains together and form a hydrogel. This strategy relies on the distribution of oppositely charged groups. Sometimes creating an ionic crosslinked hydrogel can be as straightforward as combining oppositely charged components. For example, a Ca-alginate-based hydrogel can be produced by adding calcium; the Ca^2+^ ions act as the cationic donor to carboxyl-group-rich polymers. This is a typical ionic-interaction-based self-healing hydrogel and functions well in superficial wound dressings. However, there is still a challenge for hydrogels that rely on ionic interaction: if the oppositely charged components interact too violently, an inhomogeneous substance will form. Thus, the gelation and developing strategy are critical [57].

Many of the mechanical properties, including strength, toughness, and self-healing ability, depend on the hydrogel’s molecular weight and ionic concentration; therefore, dynamic designs enable a broad spectrum of products. Further studies showed the great potential of ionic bond-based hydrogels by tuning different interactions and hierarchical structures. Sun et al. [58] presented a hydrogel with tunable toughness and viscoelasticity. Both strong and weak ionic bonds exist within the hydrogel to either keep the shape intact or enhance the resistance and self-healing. By keeping a high density of ionic bonds and a balance of different interactions, the hydrogel showed desirable mechanical properties and biocompatibility, enabling its application as a structural biomaterial. Another polyampholyte hydrogel with exquisite hierarchical structures demonstrated resistance toward fatigue fracture [47]. This study analyzed the structure of polyampholyte hydrogels and revealed the three-tier architecture (Figure 1). The reversible ionic bonds as building blocks can absorb external energy during loading. The formation of an ionic interaction also induced local aggregation of the crosslinked polymers, forming hard/soft phase networks at a larger scale and resulting in a slower advance of fatigue cracks.

#### 2.2.2. Hydrogen Bond

A hydrogen bond forms between two atoms: a hydrogen and an electronegative atom such as oxygen. It is quite common in physiological environments since an aqueous solution with organic substances provides conditions such as amide groups and hydroxyl groups, meaning that hydrogen bond-reliant hydrogels usually have good histocompatibility. Compared to covalent and ionic bonds, hydrogen bonds are much weaker; therefore, the simple application of hydrogen bonds is less competitive for uses requiring high mechanical strength [59]. However, there are still strategies to improve the performance of hydrogen bonds. The mechanical strength can be dramatically improved by applying motifs that involve multiple hydrogen bonds and other noncovalent interactions. The enhanced systems can be used for hard tissue such as bone and cartilage [59,60,61,62]. For example, Wang et al. [63] designed a multi-hydrogen bonding system based on a monomeric gelator 2-amino-2′-fluoro-2′-deoxyadenosine (2-FA) and fabricated a tough self-healing hydrogel. The multiple hydrogen bonds derived from 2′-F and double NH2 groups and water molecules ensured strength and sufficient support; with the injectability, mechanical durability, and inflammatory regulation ability, the hydrogel became a convenient and effective tooth-extraction socket healing agent.

Besides the designed supramolecule, hydrogen bonds exist in a wide range of abundant and biodegradable biopolymers. These materials can be modified to improve mechanical properties. Lu, Gu, Hu, Fu, Ye, Zhang, Zheng, Hou, Liu, and Jiang [59] proposed a biobased hydrogel that included glycerol and PVA starch, which naturally supplied the conditions of hydrogen bond formation. The hydrogel also showed high conductivity and freezing resistance and opened up the possibility of using organic hydrogels for flexible electronic devices.

#### 2.2.3. Hydrophobic Interaction

Hydrophobic bonds work by repulsion forces between hydrophobic groupings and water. In a polar solution, the hydrophobic parts of molecules spontaneously assemble to fold and form clumps due to their repulsion to water. Appel et al. [64] proposed a hydrogel designed with a hydrophobically modified polymer and NPs, and the hydrophobic interaction facilitated the self-healing of the hydrogel. With the biocompatibility of the material and erosion-based release, the hydrogel was a minimally invasive drug and cell delivery system. Other studies showed that a hydrophobic cage structure could further enhance the mechanical properties when combined with hydrogen bonds. After constructing large hydrophobic groups copolymerized with a hydrophile such as UPy motifs or acrylamide, self-healing hydrogels showed enhanced toughness [65,66].

When detergents are involved in this type of hydrogel, micelles, spheres of hydrophobic groups surrounded by amphipathic molecules, carry out the function of crosslinking backbones and enable better self-healing. Tuncaboylu [67] presented a simple strategy to produce a hydrophobic-interaction-based hydrogel model by incorporating hydrophobic sequences by micellar polymerization (Figure 2). They also optimized the length of the alkyl side chain length at 18 carbon atoms. Gulyuz and Okay [68] tested the performance of a self-healing hydrogel prepared with large hydrophobic monomer stearyl methacrylate (C18) and surfactant SDS. They found that surfactants effectively accelerate the self-healing rate and enhance the mechanical strength. The gel with SDS showed high stability and significantly improved self-healing ability compared to the virgin hydrogel.

#### 2.2.4. Metal Coordination Interactions

Metal coordination interactions form between metal cations as electron acceptors and ligands that deliver the electrons. One of the most common strategies of metal coordination for self-healing hydrogel is inspired by the adhesiveness of the mussel [40,69,70]. The coordination bonds form between Fe and catechol. The interaction can provide satisfactory mechanical strength but is still weak enough to serve as a sacrificial bond that ruptures under stress before the backbone of the hydrogel is damaged. Liang et al. [37] designed a dual-dynamic-bond hydrogel, whose Fe–catechol coordinated bond has been reported to have pH sensitivity. When combined with QCS, which has antibacterial activity, this is a good dressing for infected skin incisions and can be removed on demand [71].

Other metal coordination-based hydrogels also show multiple attractive characteristics originating from the different metal ions. Shi et al. [72] presented a hydrogel by adding silver (Ag+) ions to a solution of HA containing BP groups (HA-BP). The coordination crosslinking enables the moldability and self-healing properties of the hydrogel, making it suitable for filling irregular wound beds. Most importantly, Ag+ ions, a broad-spectrum antibacterial component, are progressively released from the hydrogel, enabling infection prevention. These properties of the hydrogel make it a ready-to-use candidate for wound healing material. Owing to the coordination potential of HA-BP, a universal strategy has been proposed that multiple common metal ions such as Ca^2+^, Mg^2+^, and Fe^2+^ can also be added to produce a restorable hydrogel [73] (Figure 3).

## 3. Clinical Application of Self-Healing Hydrogels

As mentioned before, different strategies of self-healing are available for hydrogels. With combinations of interactions that have different degrees of strength, the mechanical characteristics are variable to meet multiple purposes. For example, for cardiac tissue, the hydrogel needs to resist tension and fatigue stress; for joints, higher strength is required; and for nerve tissue, it needs to have a particular flexibility or stiffness, similar to normal tissue. When acting as a carrier, the shear-thinning of the self-healing hydrogel enables stem cells and other components to be injected and released at a stable state at the target location.

Here, we introduce some of the canonical applications of self-healing hydrogels in the medical field.

### 3.1. Wound Dressing

The disruption of skin integrity can cause severe consequences, especially for complex and chronic wounds, such as infected lesions, diabetic ulcers, and deep burns.

Skin grafting is a reliable treatment for treating refractory skin lesions. While neither auto- nor allografts are ideal due to the potential surgical trauma and rejection reaction, a wound dressing that could promote healing could be the key to filling the gap. Multiple materials have been used to design wound dressings, and hydrogels are one of the most common and promising choices due to the high biocompatibility. Recent studies have designed an injectable self-healing hydrogel with bioactive factor releasing, antibacterial and on-demand gelation, and dissolution characteristics, highlighting the utility of self-healing hydrogels in wound care [30,38,39].

Liang, Li, Huang, Yu, and Guo [37] reported a dual dynamic crosslinking self-healing hydrogel based on coordinate bonds between catechol and Fe and the Schiff bond interaction of quaternized chitosan (QCS) and protocatechualdehyde (PA). The positive component of the hydrogel could interact with and damage negatively charged bacterial structures. Moreover, the photothermal effect of Fe ions upon NIR induction can further enhance the antibacterial effect [36]. As long as the dressing needs to be removed, the application of deferoxamine mesylate (DFO), a medical Fe-chelating agent, can easily disrupt the hydrogel structure and enable on-demand removal. Another hydrogel has been designed through the supramolecular assembly of polydopamine-decorated silver nanoparticles (PDA-Ag NPs), polyaniline, and polyvinyl alcohol. Both Ag+ ions released from the hydrogel and the skin-mimicking nanostructure contributed to the antibacterial property against *E. coli* and *S. aureus*. The hydrogel was effective in diabetic foot wound infection control and collagen deposition. Furthermore, the conductive property brought in another potential function as an electrophysiological sensor [39].

Introducing antibiotics into the hydrogel sustained-release system is another strategy to achieve infection control. Wang, Wu, Long, Yang, Fu, Hu, Kong, and Wang [29] reported a pH and reactive oxygen species (ROS) dual-responsive hydrogel that contained vancomycin-conjugated silver nanoclusters and nimesulide micelles, a potent NSAID. In an inflamed wound, the low pH environment and ROS promote hydrogel transformation, which causes the release of an Ag nanocluster to enhance infection control. The further release of Nimesulide could regulate the inflammation response and promote wound healing. Furthermore, the cationic group modification of the gel and the dense mesh structure contributed to the entrapment of RBCs, promoting clotting. The highly interconnected porous structure and expansion ration also enable absorbance of blood, enabling excellent hemostatic performance. These properties make the hydrogel a potentially excellent candidate as a dressing for chronic refractory wounds.

### 3.2. Bone and Cartilage

Massive bone defects and cartilage injury are significant challenges to orthopedic surgeons. Due to the limited osteochondral regeneration function, tissue engineering became a popular option. Nowadays, standard transplant methods still cause adverse effects, including pain, rejection, and the need for additional surgery. An environment with a proper scaffold, cell matrix, and bioactive molecules is needed for bone regeneration. The hydrogel could be a candidate to replace some bone transplant treatments under certain situations with the new strategies of osteochondral tissue engineering.

Electrostatic attractions and the dynamic Schiff base reaction are common strategies for bone tissue engineering hydrogels. Recently, a catechol-conjugated chitosan (CHI-C) multifunctional hydrogel based on dynamic Schiff base linkages was reported by Huang, Cheng, Wang, Zhang, and Zhang [26], with self-healing and cytocompatibility capacity. When injected into irregular bone defects, it can heal within minutes, stop bleeding, promote bone regeneration, and be absorbed afterward. The hydrogel showed satisfying self-healing and hemostatic performance when tested in a rabbit ilium bone defect model. Zhang et al. presented another electrical interaction-based hydrogel composed of HA-BP, Ac-BP, and MgCl2. The hydrogel can also encapsulate hMSCs and promote their spreading and osteogenesis.

The dynamic osteoarticular environment naturally requires better retention and adhesion of the implants [21,40]. A nanocomposite hydrogel based on a reversible noncovalent interaction was reported to have high mechanical strength and resilience adapted to native cartilage [41]. Evaluation of the hydrogel via a bone defect animal model showed that the hydrogel could survive for as long as four weeks while aiding in the progress of bone regeneration. Yu, Cao, Du, Wang, and Chen [27] designed a self-healing hydrogel crosslinked with Diels–Alder and arylhydrazone bond. The double crosslinking ensured the structural integrity and bearing capacity. Furthermore, the aldehyde groups could interact with endogenous cartilage surface via a Schiff base bond, contributing to the solid adhesiveness of the cartilage grafts. The double network hydrogel was presented with the adhesive strength of 10.3 ± 0.7 kPa in a push out test and showed more integrated connection and no cracks to host cartilage under SEM. The property of adhesiveness was archived with the covalent bond between hydrogel and cartilage and is critical for the compatibility and stableness in cartilage defect repair.

### 3.3. Cardiac

Cell therapy is an emerging approach to improving the prognosis for treating cardiovascular diseases, especially myocardial infarction. However, the effect of direct cell injection therapy is still not ideal due to the low retention and survival rate. Hydrogel catheter delivery is a minimally invasive intervention strategy, suitable for delivering both growth factors and stem cells. Several clinical studies have shown the feasibility and safety of the therapy, while the efficacy is still under investigation [24,35,74]. In this strategy, a hydrogel’s self-healing and shear-thinning characteristics make it a great protective carrier for cell delivery and contribute to continuous retention within a mechanically dynamic environment. Additionally, the hydrogel provides a microenvironment for cell attachment and differentiation, further improving the retention of cells in cardiac tissue.

A pH-switchable hydrogel was reported, and a rapid-gelation and self-healing-enabled hydrogel could survive at the injection site [35]. Growth factors HGF/IGF-1 were successfully released and retained to form an effective intramyocardial gradient that recruited and promoted myocardium repair. A chronic myocardial infarction pig model showed that the system could be safely applied in vivo and increase blood flow to the infarcted area, resulting in a reduction of the infarct scar.

Dong et al. [24] reported a chitosan-graft-aniline tetramer (CS-AT) and dibenzaldehyde-terminated poly(ethylene glycol) (PEG-DA)-based hydrogel; with the Schiff base interaction, the hydrogel had self-healing and shear-thinning ability. As the benefits of electroactive and conductivity for electrical stimuli sensitive cells such as myocardial cells has been well demonstrated, electrophysiological property is also an aspect of biocompatibility. The choice of CS-AT polymer was made due to its conductivity brought by aniline tetramer. According to the research, myoblasts and adipose-derived mesenchymal stem cells experienced no significant decrease in viability after being incubated in the hydrogel. The cells could be successfully released in vivo at a tunable rate by adjusting the concentration. Additionally, the conductivity of the hydrogel being similar to that of the myocardium allowed for the maintenance of the electrophysiological environment and promoted cardiac myocytes’ regeneration.

Small extracellular vesicles (sEV) derived from mesenchymal stem cells are another promising bioactive factor that could be used in non-cell systems for post-MI treatment. Lv et al. [75] tested bone marrow mesenchymal stem cell (MSC)-derived small extracellular vesicles incorporated in alginate hydrogel, which proved to be biocompatible and effective in some clinical trials. The sEVs-embedded self-healing hydrogel had a better retention rate compared to injection only, which led to angiogenesis and inflammation suppression.

### 3.4. Neural Injury

Tissue engineering has become a key topic in relation to neural injury treatment. The use of biocompatibility scaffolds provides a protective environment for neuron regeneration. Like cardiac tissue engineering, hydrogels effectively alleviated the damage caused by shearing force when injected to support injured tissue. Due to the particularity of nervous tissue, the material needs to have similar toughness, long retention time, and proper electroconductibility. Nowadays, self-healing hydrogels are under investigation in different scenarios of neural injuries.

CNS injury is a critical clinical issue, but few therapies show remarkable effects. Stem cell therapy is a promising strategy for neural regeneration, and injectable hydrogels are a particularly appealing delivery system. Tseng, Tao, Hsieh, Wei, Chiu, and Hsu [25] proposed a self-healing hydrogel based on a Schiff base interaction between chitosan and DF-PEG. By adjusting the concentration of the components, the stiffness of the gels can be adjusted. The experiment with a zebrafish model demonstrated an appropriate stiffness of 1.5 kPa that effectively supported the injury area without disrupting regenerated structures. Moreover, the effectiveness of the hydrogel was verified with an EtOH-induced brain injury model; zebrafish treated with a chitosan-based self-healing hydrogel encapsulating neuroprogenitors had the best recovery rate.

As for spinal cord injury, the hydrogel can also provide a protective environment that attenuates inflammation and promotes axon growth. A recent study reported on a hydrogel fabricated with N-terminus with fluorenylmethoxycarbonyl (Fmoc) grafted chitosan and peptide with curcumin encapsulated. The self-healing property was achieved by stacking and hydrophobic interaction, which also ensured enhanced durability and a sustained release of curcumin. This multifunctional hydrogel system is able to induce Schwann cells’ recruitment and remyelination, further promoting the recovery of walking function and making promising progress toward repairing spinal cord injury when tested with a rat spinal transection model [34].

Besides injectable hydrogels, self-healing hydrogels can be applied in other forms, such as a thin film-like dressing for injured nerve fibers. In one study, the hydrogel was composed of tannic acid and PPy. Compared to traditional conductive conduits, it was soft, plastic, and able to curl around the target nerve fiber automatically. The highly aligned porous microstructures and interfacial interaction promoted cell attachment such that Schwann cells could spread out in good stretching state with strong interconnectivity and distribute to the denuded area. The biocompatible electroconductive hydrogel led to significant improvement in nerve regeneration and muscle function retention in peripheral nerve-injured animals [31].

### 3.5. Cancer Therapy

The treatment of cancer has always been a research focus. Due to the heterogeneity and resistance of tumor cells, smart medication and combined therapy have become a meaningful research direction.

Cancer cells are more sensitive to short-term high-dose exposures to chemotherapeutic drugs than to constant concentrations. Thus, precisely targeted drug release seems to be a plausible strategy [23,42,43,76]. Huebsch, Kearney, Zhao, Kim, Cezar, Suo, and Mooney [43] proposed a hydrogel system whose self-healing characteristics would allow for repeated reversible disruption and on-demand ultrasound-induced drug release. An experiment to treat xenograft tumor models showed a substantial reduction in tumor growth, demonstrating it as a potential therapeutic application. This system can also be applied with other existing drugs and improves their performance in situations in which a “burst” of drug release is favored.

Guedes, Wang, Fontana, Figueiredo, Lindén, Correia, Pinto, Hietala, Sousa, and Santos [42] reported an injectable self-healing hydrogel prepared for chemo/photothermal therapy. The hydrogel was constructed with dual crosslinking by Schiff base and electrostatic interaction, incorporating Mo_154_ and doxorubicin. The lower pH in the tumor environment can trigger the release of chemotherapy drugs. Another stimulus is near-infrared light (NIR); Mo_154_ enables a rapid temperature increase under laser irradiation, inducing tumor ablation and further increasing the release rate of doxorubicin. Applying a dual-function hydrogel led to much stronger tumor suppression in an animal model.

Interestingly, Zhao, Feng, Liu, Tang, Du, Ji, Xie, Zhao, Wang, and Chen [32] put forward a guanosine-based hydrogel, whose main component, isoguanosine–borate–guanosine (isoGBG), has antitumor properties, inducing tumor cell apoptosis. Thus, this system has a dual antitumor function, integrating anticancer drugs and serving as a chemotherapeutics carrier. It also models the possibility of incorporating different functional supramolecules to design other new drug delivery systems.

## 4. Conclusions and Discussion

A hydrogel is a water-absorbing material formed after the crosslinking of polymer monomers, in which a 3D network structure is formed to contain water of different phases. After hydration, hydrogels have a softness close to human tissue and can be degraded into innocuous metabolites, which also contributes to their better biocompatibility over other materials [13,22,77,78]. These unique characteristics make it valuable for regenerative medicine applications. Despite the advantages, hydrogels are usually fragile and not durable. They could be used in practical applications such as cell culture scaffolds, but there are still challenges to clinical use [4,5].

There are two strategies for self-healing materials: either rely on reversible interactions or add capsules containing analogous materials that can be released to repair the defect. As capsules in the second strategy may affect the structure of the material and limit self-healing, self-healing materials that apply dynamic interactions have gained increasing attention. Multiple types of reversible interactions can be used to design self-healing hydrogels, and variable concentrations and strengths of bonds contribute to the unique physicochemical properties of the hydrogel [2,77]. Here, we have mainly summarized the mechanisms of self-healing hydrogels based on reversible interactions and discussed their applications in the medical field.

The improving technologies and strategies allow researchers to develop medical applications in a variety of ways from delivery system to regenerative implants. We reviewed several typical utilizations of self-healing hydrogel in tissue repairment and molecule/cell delivery at the skin, bone, cardiac, and neural system. Though most of the studies are still at an early stage, the variety of choice and current promising outcomes shed a light on the great field of medicine advancements. The significance of self-healing hydrogels in the medical field is reflected not only in their high durability and spontaneous repair ability but also in their responsiveness and flexibility. By modifying the polymer and introducing other active ingredients, hydrogels can present physicochemical characteristics suitable to different environments.

On the one hand, hydrogels have outstanding mechanical strength and fatigue resistance, which endow the implants with longer lifetimes; involve less consumption; and, above all, create less collateral damage caused by repeated replacement. On the other hand, the microscopic three-dimensional structure and reversible interaction force of hydrogels give the material greater potential to be modified and have space for function-related substrates including cells, bioactive factors, antibiotics, chemotherapy drugs, etc. Stem cell therapies or replacing differentiated tissues or whole organs are two major strategies for regenerative medicine, in which self-healing hydrogels can contribute to maintaining cell activity and present as scaffolds or carriers that minimally disrupt normal biological processes.

Although significant advances have been made in the discovery and application of self-healing hydrogels, many challenges remain to be addressed. More delicate experiments are needed to ensure the stability and reliable quality of the hydrogel, especially for in vivo implants in critical organs, such as the brain and heart, and when carrying toxic chemotherapy drugs, in which any accidental displacement or rupture could have grave consequences. Additionally, self-healing hydrogels are difficult to prepare and are not ready for mass production. For hydrogels embedded with active substances and living cells, preservation is another challenge to large-scale clinical application. Overall, research into self-healing hydrogels in the medical field is still in the initial stages, and there are still many problems and deficiencies. Although most studies are still in the early stages, they have offered important possibilities. Self-healing hydrogels can be used not only as implants but also as carriers. In the current era of the rapid development of biologics and regenerative medicine, this application shows great potential. We believe that, if solutions can be found to these problems, the application of self-healing hydrogels in clinical medicine will bring about great developments in the field.

## Figures and Tables

**Figure 1 materials-16-01215-f001:**
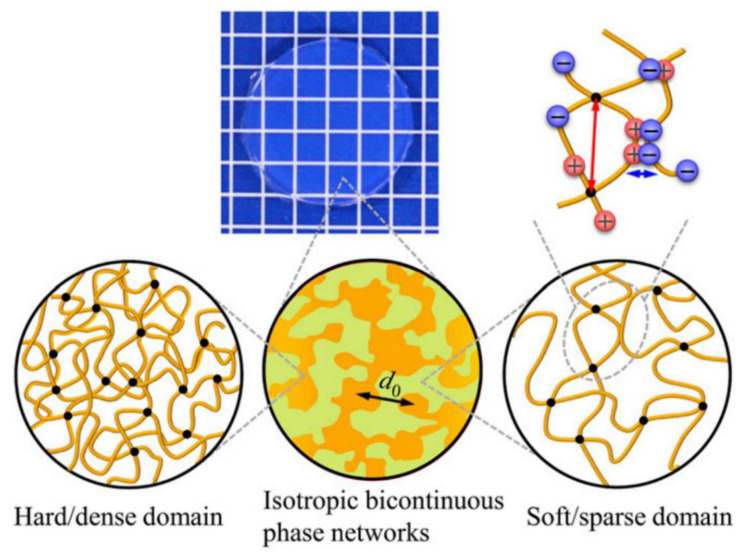
Ionic bonds at the 1 nm scale (blue arrow), cross-linked polymer network at the 10 nm scale (red arrow) and bicontinuous hard/soft phase networks at d0~100 nm scale formed the hierarchical structure of PA gel (reprinted with permission of PNAS, from [47]).

**Figure 2 materials-16-01215-f002:**
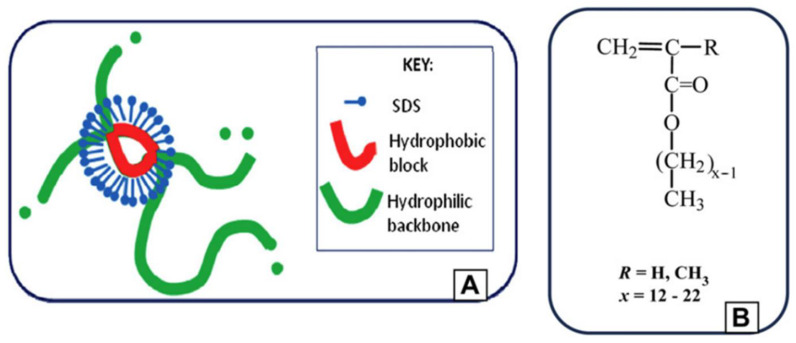
Self-healing gel (**A**) and the hydrophobic monomer (**B**). (Reprinted with the permission of Elsevier from [67]).

**Figure 3 materials-16-01215-f003:**
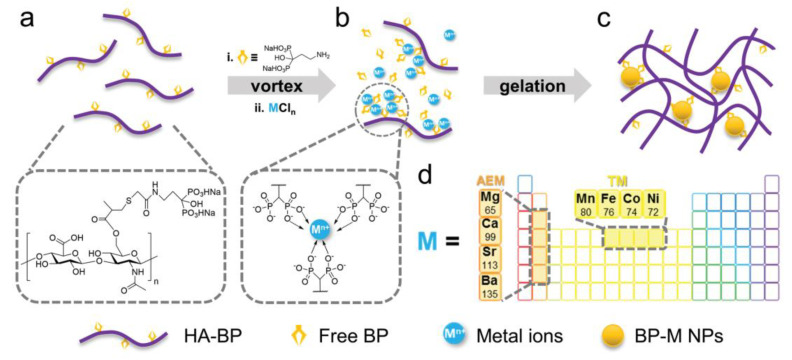
The structure of self-assembled HA-BP-M nanocomposite hydrogels (**a**–**c**) and representative elements (**d**). (Reprinted with permission of John Wiley and Sons from [73]).

**Table 1 materials-16-01215-t001:** Clinical applications of self-healing hydrogels: a brief summary of recent studies.

Self-Healing Mechanism	Material	Publication Year	Animal Model	Application	Ref.
Schiff base bond	N,O-carboxymethyl chitosan (N,O-CMCS)-guar gum-base	2021		anticancer drug delivery	[23]
	chitosan-aniline tetramer (CS-AT) and PEG double aldehyde PEG-DA	2016		cardiac cell therapy	[24]
	glycol chitosan and telechelic difunctional poly(ethylene glycol) (DF-PEG)	2015	zebrafish embryo neural injury model	CNS injury treatment	[25]
	Catechol-conjugated chitosan (CHI-C) and dialdehyde cellulose nanocrystal (DACNC)	2021	mouse liver injury model, mouse tail amputation model, and rabbit ilium bone defect model.	bone defect treatment	[26]
Diels–Alder bond	HA, furylamine amine groups and adipic dihydrazide (ADH)	2015		cartilage engineering	[27]
Coordination interaction	HA-BP, acrylated bisphosphonate (Ac-BP), and MgCl_2_	2017		osteogenesis stem cell therapy	[28]
	3-carboxy-phenylboronic acid, gelatin, and vancomycin-conjugated silver nanoclusters	2021	mouse hemorrhaging liver model, chronically infected wound in a diabetic mouse model	wound treatment	[29]
	hyaluronic acid-graft-dopamine (HA-DA) and reduced graphene oxide (rGO)	2019	mouse full-thickness wound model	wound treatment	[30]
	tannic acid, TA, PPy, and Fe3+	2021	diabetic sciatic nerve injury model	peripheral nerve injury	[31]
	isoguanosine-borate-guanosine (isoGBG)	2020	OSCC xenograft mouse model	cancer therapy	[32]
Dynamic imine bond	glycol chitosan and poly(N-isopropylacrylamide)-co-poly(acrylic acid) (DF poly(NIPAM-co-AA))	2016		drug delivery and 3D cell cultivation	[33]
Π–π stacking interactions	Fmoc-grafted chitosan (FC) and Fmoc peptidelaminin-derived peptide IKVAV (FI)	2021	spinal cord transection rat model	spinal cord injury treatment	[34]
Hydrogen bonds	supramolecular ureido-pyrimidinone (UPy) and (PEG) chains	2013	chronic myocardial infarction pig model	infarcted myocardium treatment	[35]
Hydrogen bonding, hydrophobic and coordination interaction	poly(glycerol sebacate)-co-poly(ethylene glycol) (PEGSD)and UPy-HDI synthon modified gelatin (GTU)	2020	rabbit ear artery bleeding model; full-thickness rat skin incision model	wound treatment	[36]
Schiff base and coordination interaction	(Fe), protocatechualdehyde (PA) and quaternized chitosan (QCS)	2021	rat skin incision model and infected full-thickness skin wound model		[37]
	aminated gelatin (AG)-oxidized hyaluronic acid (OD-Fe(III))	2021	rat femoral vein puncture model, hemorrhaging liver model and jugular vein embolization model	hemostasis wound treatment	[38]
Coordination interaction and hydrogen bonds and π–π stacking	polydopamine decorated silver nanoparticles (PDA-Ag NPs) and polyaniline and polyvinyl alcohol (CPHs)	2019	diabetic rat model, infected full-thickness skin wound model	epidermal sensor and diabetic foot wound treatment	[39]
Coordination interaction and disulfide bonds	adhesive liposomes—polyethylene glycol (A-LIP- PEG) and	2019	osteoporosis rat model	bone reconstruction treatment	[40]
Electrostatic and hydrophobic interaction	HABP·CaP hybrid nanocomposites	2014	bilateral frontal femoral condyle defect model	bone reconstruction treatment	[41]
Dynamic imine bond and electrostatic interactions	DF-PEG and chitosan (CS-g-PNIPAAm)	2021	murine melanoma model (B16.OVA)	chemo–photothermal therapy	[42]
	calcium alginate hydrogels	2014	xenograft breast tumors model	intermittent high pulsing drug release	[43]
Hydrophobic interactions and hydrogen bonds	acryloyl-6-aminocaproic (acidAA) and AA-g-N-hydroxysuccinimide(AA-NHS)	2021	swine gastric wound model	gastric wound treatment	[44]
	gellan gum (GG)	2019	rat corneal injury model	corneal sheet dressings	[44]

## Data Availability

Not applicable.

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
