# Peer review of "Advances and Progress in Self-Healing Hydrogel and Its Application in Regenerative Medicine"

_materials, 2023, doi:10.3390/ma16031215_

Round 1
Reviewer 1 Report
The title of the review is “advances and progress in self-healing hydrogel and its application in regenerative medicine”. The content of the review article is very poor since the authors did not exhaustively review the subject. For example, the authors only cited 50 articles that do not represent the state of the art on the issue. Also, the authors require to restructure the content of the review.
Reviewer comments: The number of new publications in the field is high and growing day by day. But it is also true that the number of good reviews in the area is equally increasing. Hence, I believe new reviews should be focused on the recent advances while making use of the efforts from previous reviews to substantiate the knowledge in the field. The authors can add a statistical graph of the articles published in the last 10-12 years, referring to self-healing hydrogel and its application in regenerative medicine.
Reviewer comments: It is still difficult to find the novelty of the work concerning what has already been published. A literature review is required. What is the difference between what is published with what the authors want to publish? It is not clear. The authors must describe these differences.
Reviewer´s comment: The authors can improve the definition of hydrogels, considering all their fascinated properties. The authors can cite: https://doi.org/10.1016/j.eurpolymj.2020.110176
Reviewer´s comment: The authors must add a section dedicated to regenerative medicine and describe the current challenges.
Reviewer´s comment: What are the main physicochemical, mechanical, and biological properties that must possess a hydrogel for regenerative medicine. Explain more in respect in the section.
Reviewer´s comment: The authors must add a section that describes the polymers more used to produce self-healing hydrogels.
Reviewer comments: The authors should better describe the research they place in the review.
Reviewer comments: In the Clinical application of the Self-healing hydrogel section, the authors should describe the main articles related to the issue. However, the articles that are described are very few, and the descriptions are very poor.
Reviewer´s comment: Conclusions and perspectives??????
Reviewer 2 Report
Recently, the properties of hydrogels and their numerous applications have received more and more attention. In particular, the manifold applications of hydrogels in the field of biomedicine, primarily the regenerative one, are of great interest. In this regard, the present review article makes a certain contribution to the generalization of knowledge about these extremely interesting materials.
The article can be published after the authors cover the following aspects in detail in the article. First of all, in the given medical applications of hydrogels, the authors need to explain in detail the principles of SELF-healing mechanisms, and not just list the various areas of hydrogel usage. In my opinion, it is necessary to strengthen the emphasis dedicated specifically to self-acting of hydrogel repair after external damage. Second, the question on self-healing action zones in the volume of hydrogels remained completely unsolved. Hydrogels are a three-dimensional supramolecular system whose properties are generally manifested on a macroscopic scale. The authors review methods for modifying the volume properties of hydrogels. And what happens if external damage affects only a certain part of the hydrogel volume? What approaches are used in such case?
Reviewer 3 Report
The manuscript by Zhu et al. summarizes the design options to form self-healing hydrogels and their applications in tissue engineering, focus on clinical practice. It is strongly recommended to include the following observations:
Comments to addresses
1. It is recommended to include the definition of self-healing hydrogels in the abstract, as well as improve the connection between the ideas.
2. The following idea (line 26) is not clear:
The interaction between hydrogels counter-balanced leading to expansion of polymer 26 networks, which also regulates hydrogels’ internal transport, diffusion characteristics, 27 and mechanical strength
3. It is recommended to improve the connection between ideas that are described in the introduction. For example, there is not a clear association between the ideas exposed in the second, third and fourth paragraphs.
4. It is recommended to include a diagram to show the different reactions that are explained in section 2.1, for example, show how the Schiff base bond is formed.
5. Revise Figure 1. The legend is misplaced. Similarly, please, improve resolution of Figure 3.
6. Section 2 includes different investigations that are used to explain how covalent and noncovalent bonds are utilized to form self-healing hydrogels. However, this section lacks detail. It is suggested to include a summary table in which the authors report a number of studies that support this section 2.
7. Table 1 title is misplaced.
8. In table 1, please, include the chemical groups that participate in the bonds that make the hydrogel being self-healing. Also, it is recommended to include the most relevant results or findings for each study. Also, organize the information by year, or by bond type, or by application.
9. Sections 3.3, 3.4 and 3.5 might high potential to give some examples of self-healing hydrogels used in different tissues. However, the mechanism of action of the hydrogel systems that are described, is not completely clear.
10. Revise spelling of the title of section 4
Round 2
Reviewer 1 Report
The authors did not properly attend to the comments since they were too short. The article is not of sufficient quality to be published in this journal. The content of the review article is very poor since the authors did not exhaustively review the subject. Also, the authors require to restructure the content of the review.
Point 1: The number of new publications in the field is high and growing day by day. But it is also true that the number of good reviews in the area is equally increasing. Hence, I believe new reviews should be focused on the recent advances while making use of the efforts from previous reviews to substantiate the knowledge in the field. The authors can add a statistical graph of the articles published in the last 10-12 years, referring to self-healing hydrogel and its application in regenerative medicine.
Response 1: We mainly reviewed recent studies within 3 years and focused on how we can take advantage of the characteristics of self-healing hydrogels in the medical field. We also added a table to help summarize the examples of application directions.
[second review] Thank you for the changes made. However, the authors must clarify what do they mean by “characteristics of self-healing hydrogels”? Also, what table number are you referring to?
Point 2: It is still difficult to find the novelty of the work concerning what has already been published. A literature review is required. What is the difference between what is published with what the authors want to publish? It is not clear. The authors must describe these differences.
Response 2: We mainly focused on recent studies and discussed the features of self-healing hydrogels that are suitable for medical application and their potential benefits from a clinical perspective. We believe that self-healing hydrogel is actually a collection of materials with different characteristics which were endowed with the diversity of backbone polymers, chemical interactions and self-healing strategies.
[second review] Thank you for the changes made. However, the authors did not address this question. The novelty of the work is not yet clear, since they do not discuss the articles previously published. The authors must answer: What is the difference between what is published with what the authors want to publish? It is not clear. The authors must describe these differences.
Point 3: The authors can improve the definition of hydrogels, considering all their fascinated properties. The authors can cite: https://doi.org/10.1016/j.eurpolymj.2020.110176
Response 3: We now add clarification of the hydrogels and cited the recommended research started from line53.
[second review] Thank you for the changes made. However, the authors did not improve the definition of hydrogels.
Point 4: The authors must add a section dedicated to regenerative medicine and describe the current challenges.
[second review] Thank you for the changes made. However, the authors did not address this question. The review title is: “Advances and progress in self-healing hydrogel and its application in regenerative medicine”. So clearly, the authors must describe regenerative medicine's generality and the current challenges, where self-healing hydrogels help solve them.
Response 4: We add descriptions of the circumstances of regenerative medicine from line 254-263 and discussed the challenges from line 458.
Point 5: What are the main physicochemical, mechanical, and biological properties that must possess a hydrogel for regenerative medicine. Explain more in respect in the section.
Response 5: We addressed the physicochemical properties of the self-healing hydrogel and its significance for clinical application from line 254. And we also described how self-healing and other properties are achieved and promoted their medical application.
[second review] Thank you for the changes made. The authors' response is very short. In the literature, there is a lot of information on this topic, so the authors can discuss more in-depth, describing and discussing examples with numerical values. I don't think that mechanical properties alone are the only valuable properties for clinical applications: where is the porosity, pore size, interconnectivity, biocompatibility, surface area etc, etc,????? It seems that the authors took the comments made by the reviewers as a bad joke. They must take them seriously.
Point 6: The authors must add a section that describes the polymers more used to produce self-healing hydrogels.
Response 6: We added introductions to the understanding of polymers used in self-healing hydrogels from line 53-59, and we also included the information of polymers when describing different hydrogels.
[second review] Thank you for the changes made. The authors' response is very short. More information is required.
Point 7: The authors should better describe the research they place in the review.
Response 7: We include more descriptions of the research. For example, from line 370-390, we added more detailed introductions of the results of the study.
[second review] Thank you for the changes made. However, the authors must discuss the articles cited more in detail.
Point 8: Clinical application of the Self-healing hydrogel section, the authors should describe the main articles related to the issue. However, the articles that are described are very few, and the descriptions are very poor
Response 8: We addressed the related question from line 254-262 and discussed the medical application in discussion from line4580. The self-healing ability is the key to the favorable features of hydrogels including the enhancement of mechanical strength, flexibility and durability, and injectability, and we also discussed how these factors contribute to the performance in clinical application.
[second review] Thank you for the changes made.
Point 9: Conclusions and perspectives??????
Response 9: We now add our understanding and discussion of the topic from line 426 to 472
[second review] Thank you for the changes made. However, the conclusion and discussion sections do not go together.
Reviewer 2 Report
I agree with corrections. Manuscript can be accepted in the present form.
Author Response
We gratefully appreciate your valuable suggestions.
Reviewer 3 Report
Thank you for answering my questions. However, the authors still need to clarify the following comments:
1. Section 2 includes different investigations that are used to explain how covalent and noncovalent bonds are utilized to form self-healing hydrogels. However, this section lacks detail. It is suggested to include a summary table in which the authors report a number of studies that support this section 2.
The manuscript does not address this comment.
2. In table 1, please, include the chemical groups that participate in the bonds that make the hydrogel being self-healing. Also, it is recommended to include the most relevant results or findings for each study. Also, organize the information by year, or by bond type, or by application.
The authors did not address this comment.
3. Sections 3.3, 3.4 and 3.5 might high potential to give some examples of self-healing hydrogels used in different tissues. However, the mechanism of action of the hydrogel systems that are described, is not completely clear.
The mechanisms of action are not clear, yet.
4. It is strongly recommended to improve the quality of the figures.
Author Response
Reply:
1. Section 2 includes different investigations that are used to explain how covalent and noncovalent bonds are utilized to form self-healing hydrogels. However, this section lacks detail. It is suggested to include a summary table in which the authors report a number of studies that support this section 2.
The manuscript does not address this comment.
Answer: Thank you for your advice. In manuscript, we have added the table 1 including covalent and noncovalent bonds that are utilized to form self-healing hydrogels and its details like authors and applications.
2. In table 1, please, include the chemical groups that participate in the bonds that make the hydrogel being self-healing. Also, it is recommended to include the most relevant results or findings for each study. Also, organize the information by year, or by bond type, or by application.
The authors did not address this comment.
Answer: Thank you for your advice. We have rearranged the Table 1 by bond type. The table 1 included the most relevant results or findings for each study.
3. Sections 3.3, 3.4 and 3.5 might high potential to give some examples of self-healing hydrogels used in different tissues. However, the mechanism of action of the hydrogel systems that are described, is not completely clear.
The mechanisms of action are not clear, yet.
Answer: We have revised the content in section 3.3, 3.4 and 3.5. line 393-395, line 412-414, line 430-436, line 440-443, line 459-463, and line 468-472.
4. It is strongly recommended to improve the quality of the figures.
Answer: Thank you for your advice. We have improved the quality of the figures.

Round 3
Reviewer 1 Report
It is still difficult to find the novelty of the work concerning what has already been published. A literature review is required. What is the difference between what is published with what the authors want to publish? It is not clear. The authors must describe these differences in the introduction section.
